# Friend or Foe: The Role of the Cytoskeleton in Influenza A Virus Assembly

**DOI:** 10.3390/v11010046

**Published:** 2019-01-10

**Authors:** Sukhmani Bedi, Akira Ono

**Affiliations:** Department of Microbiology and Immunology, University of Michigan, Ann Arbor, MI 48109, USA; sukhbedi@umich.edu

**Keywords:** influenza, cytoskeleton, actin, microtubules, virus assembly

## Abstract

Influenza A Virus (IAV) is a respiratory virus that causes seasonal outbreaks annually and pandemics occasionally. The main targets of the virus are epithelial cells in the respiratory tract. Like many other viruses, IAV employs the host cell’s machinery to enter cells, synthesize new genomes and viral proteins, and assemble new virus particles. The cytoskeletal system is a major cellular machinery, which IAV exploits for its entry to and exit from the cell. However, in some cases, the cytoskeleton has a negative impact on efficient IAV growth. In this review, we highlight the role of cytoskeletal elements in cellular processes that are utilized by IAV in the host cell. We further provide an in-depth summary of the current literature on the roles the cytoskeleton plays in regulating specific steps during the assembly of progeny IAV particles.

## 1. IAV life cycle

IAV is an enveloped virus with an eight-segmented negative sense RNA genome. On the surface of the virus are three glycoproteins: hemagglutinin (HA), neuraminidase (NA), and the ion channel protein (M2). Immediately below the viral membrane, the matrix (M1) protein oligomerizes to coat the inside of the virus particle [1]. The viral RNA segments incorporated into the virus particle are present as a viral ribonucleoprotein (vRNP) complex between the viral RNA, nucleoprotein (NP), and three polymerase subunits: PA, PB1, and PB2 [2]. For entry, IAV first binds to the target cell via interactions between HA on the viral membrane and sialic acids on the host cell membrane [3,4]. Following this interaction, the virus is internalized via the host cell’s endocytic pathway. The acidic pH of the endosome causes a conformational change in HA, exposing a fusion peptide that allows the fusion of the endosomal membrane with the viral membrane [5], following which the vRNPs can be released in the cytosol and imported into the nucleus. Viral RNA replication and transcription of mRNAs take place in the nucleus, followed by viral protein translation in the cytoplasm. Assembly of new virus particles requires the trafficking of at least four viral structural proteins, HA, NA, M1, and M2, as well as vRNPs, to the plasma membrane [6,7]. After incorporation of the viral genome into budding virus particles, the M2 protein mediates scission and thus release of the nascent particles from the cell [8].

## 2. Overview of the Cytoskeleton

The dynamic nature of the cytoskeleton allows for the regulation of a wide range of cellular functions [9,10]. The cytoskeleton is comprised of three main cytoskeletal components: microtubules, actin filaments, and intermediate filaments. All three form filament-like structures that are organized to carry out three main functions: spatial organization within the cell; connecting the cell to the external environment; and regulation of cell motility and shape.

Microtubules self-assemble from tubulin dimers and form long filaments, which function in chromosome segregation during cell division, the transport of cargo such as intracellular vesicles and organelles, and the maintenance of cell polarity [11,12]. They are in most, but not all, cell types anchored to an organizing center, i.e., a centrosome in most cases, in the cell interior, and radiate outwards to the periphery. The minus end of the microtubule is anchored to the centrosome and the growth and shrinkage of the polymer takes place at the plus end. Two motors, dynein and kinesin, associate with microtubules and drive the transport of intracellular cargo towards and away from the center of the cell, respectively [13].

Actin monomers (G-actin) polymerize to form double-helical filaments (F-actin), which are thinner and less stiff than microtubules [10,14]. Actin filaments are asymmetric in nature, with a barbed end that grows faster and a pointed end that loses actin monomers faster. Unlike microtubules, actin filaments can form more complex structures, such as branched networks, bundled networks, and non-aligned networks. These different conformations of F-actin require different actin-nucleating proteins [15,16]. While the Arp2/3 complex nucleates branched F-actin structures, formins nucleate unbranched actin filaments and serve as elongation factors for the growing actin filament. In addition to nucleating factors, the organization of F-actin is dependent on cofilin, an actin-binding protein that severs filaments to generate free ends for the addition of G-actin [17]. Additionally, unlike microtubules, actin filaments do not grow outwards from an organizing center, but polymerize and depolymerize locally in response to different stimuli. The actin cytoskeleton functions in cell migration, providing the structure and strength of the plasma membrane, trafficking of cargo within the cell, organization of organelles, and regulation of cytokinesis during cell division. These functions are carried out with the help of myosin motors, which play key roles in regulating the dynamics of F-actin networks [18]. The Rho family of small GTPases also regulates actin assembly and disassembly. The most well-characterized Rho GTPases are RhoA, Rac, and Cdc42. RhoA stimulates the activity of formins and promotes the assembly of linear F-actin bundles. On the other hand, Rac and Cdc42 promote branching of the actin cytoskeleton by activating the Arp2/3 complex [19]. The activity of Rho GTPases is also regulated by microtubules and in turn, they exert their effect on the microtubule ends and fine-tune microtubule dynamics [20]. Therefore, their influence on F-actin and microtubule dynamics, as well as their regulation by microtubules, also make Rho GTPases important players in the crosstalk between these two cytoskeletal elements.

Intermediate filaments are the least stiff of the three major cytoskeletal elements. They are cross-linked to each other, as well as to actin filaments and microtubules, and help in resisting tensile and shear forces. Unlike microtubules and actin filaments, intermediate filaments do not support the movement of molecular motors [21,22]. More recently, a fourth group of cytoskeletal elements, septins, has been found in most eukaryotic cells [23,24,25]. Septins bind directly to membranes and polymerize into filaments, which allows them to help organize cell membranes [26]. In the context of IAV assembly, roles for microtubules and the actin cytoskeleton have been relatively well-described. Therefore, we focus on these cytoskeletal elements in this review.

## 3. Cytoskeletal Functions Relevant to IAV Infection

This section describes the roles of the actin cytoskeleton and microtubules in driving cellular processes relevant to IAV infection, in particular the processes implicated in the latter half of the IAV life cycle.

### 3.1. Endocytic Transport

Endocytic trafficking within the cell plays an important role in many cellular functions, such as receptor signaling, cell adhesion and migration, membrane protein turnover, and nutrient uptake. Trafficking of endosomes is highly dependent on the actin cytoskeleton and microtubules, as well as the motors that associate with them [27]. The first step in the endocytic pathway involves the uptake of material from the plasma membrane, which mainly occurs via clathrin- or non-clathrin-mediated endocytosis, macropinocytosis, and phagocytosis [28]. F-actin plays an important role in remodeling the membrane and providing support to stabilize and elongate the newly formed vesicle during these early steps [29,30,31]. The myosin VI motor is involved in the transport of vesicles away from the cell cortex [32,33], following which the cargo switches over to microtubule-based movement. After this switch, the dynein motor drives the trafficking of cargo towards the interior of the cell, where the cargo is sorted for delivery to different organelles [34,35].

IAV harnesses the endocytic pathway for two crucial steps in its lifecycle: genome entry into the target cell and genome packaging into assembling virions. Genome entry is mediated by either endocytosis (clathrin-dependent or clathrin- and caveolin- independent pathways) [36] or by macropinocytosis [37], depending on the size of the entering virus particle. Trafficking of the newly replicated genome to the plasma membrane and its packaging into assembling virus particles is thought to be mediated by recycling endosomes [38,39,40]. In the cell, recycling to the plasma membrane may be “fast” or “slow”. Fast recycling is regulated by the GTPases Rab4 and Rab35, while slow recycling is mediated by Rab11 [41]. Rab11 also localizes in the Golgi and post-Golgi vesicles and may serve as a link between the endocytic and exocytic pathways [42]. The ubiquitous localization of Rab11 in the cell also makes it difficult to understand the exact nature of the recycling compartment in the cell. Currently, it is known that the slow recycling pathway involves the formation of an endocytic recycling compartment (ERC), which is a collection of tubular structures associated with microtubules. Depending on the cell, the ERC may be condensed around the microtubule-organizing center (MTOC) or be more dispersed in the cytoplasm. The dynein motors support the sorting of cargo into the ERC, while the kinesin motors drive the movement of ERC-derived vesicles to the cell periphery [43,44,45]. The budding of ERC-derived vesicles is also assisted by the actin cytoskeleton [46]. In addition, once cargo travels via Rab11-positive vesicles and reaches the cell periphery, it is transported to the plasma membrane by F-actin and its associating motors, mainly myosin Vb, which associates with Rab11 and other proteins associated with these vesicles [47,48]. Overall, the cytoskeleton and the motors that associate with it play a crucial role in organizing and driving the transport of cargo via endocytic pathways in the cell.

### 3.2. ER-Golgi Transport

The IAV transmembrane proteins HA, NA, and M2 traffic from the ER to the plasma membrane via the anterograde transport pathway. The distribution of the ER within the cell is regulated by the actin cytoskeleton and microtubules [49]. After synthesis, translocation, and initial modifications at the ER, secretory and transmembrane proteins are packaged into vesicles, which are then transported to the *cis* Golgi. These ER-derived vesicles move along microtubules just like endosomes using the dynein motor [50]. A crosstalk between the actin cytoskeleton and microtubules is also known to be important for the ER-to-Golgi transport [51]. These cargo-carrying vesicles fuse with the *cis* Golgi, after which the proteins are further post-translationally modified and sorted for delivery. F-actin and microtubules associate with the Golgi and are thought to be important for its structural integrity [49]. The cargo processed in the Golgi exits the *trans* Golgi network in vesicles, which fuse with the plasma membrane or other organelle membranes. In the case of trafficking to the plasma membrane, these vesicles move along microtubules using kinesin motors [52]. In addition, F-actin associates with post-Golgi vesicles via the actin-binding protein cortactin and supports their transport [53]. Altogether, the current literature highlights the important role that the actin cytoskeleton and microtubules play in the transport of proteins via the ER-Golgi pathway.

### 3.3. Maintenance of Cell Polarity

Differentiated epithelial cells such as those lining the respiratory tract are tightly packed to form a monolayer, which is organized and strengthened by cell-to-cell adhesion and the maintenance of cell polarity. Cell polarity manifests in the form of the asymmetric distribution of cellular components, such as trafficking cargo, plasma membrane-associated proteins, organelles, and the cytoskeleton between the apical and basolateral sides of the cell [54]. Viruses that infect polarized cells (such as IAV) utilize this polarity and restrict their entry and subsequent spread to either, but not both, sides of the cell [55].

The cytoskeleton plays a key role in the creation and maintenance of cell polarity [54]. In epithelial cells, microtubules are not anchored to the centrosome, but are organized in parallel arrays along the apico-basolateral axis, with plus ends enriched close to the basolateral side and minus ends enriched close to the apical side [56]. Consequently, microtubules play a key role in the sorting and selective delivery of proteins to the apical and basolateral surfaces [57,58,59], a process that is important for maintaining epithelial cell polarity. Selective delivery of IAV structural components to the apical membrane and the role of microtubules in this process are discussed below. In the case of the actin cytoskeleton, F-actin and actin-binding proteins are non-uniformly distributed in polarized cells, with the apical side being more heavily enriched than the basolateral side [60,61]. Since IAV encounters this apical actin network during its entry and assembly in epithelial cells (discussed below and [62]), the actin cytoskeleton could regulate these stages of the IAV life cycle via its role in maintaining the polarity of epithelial cells.

### 3.4. Organization and Maintenance of Plasma Membrane Microdomains

The plasma membrane is heterogeneous in composition and fluidic in nature. It is composed of discrete, but often dynamic, domains, with unique physical and biological properties. These specialized regions are referred to as microdomains. Of these, the most extensively studied are lipid rafts or membrane rafts, which are enriched in cholesterol and sphingolipids [63,64]. The smallest microdomains span a 10-nm diameter, but merge together to form larger domains that are several micrometers in diameter. Larger micron-size microdomains are assembled in cells in response to activation stimuli, such as in the case of receptor engagement. While lipid compositions were initially thought to be the main determinant for the formation and stability of the microdomains, it has been suggested more recently that membrane-associated proteins also play important roles in the formation and maintenance of plasma membrane microdomains [64]. IAV structural proteins also localize to these microdomains during viral assembly and drive their coalescence into larger domains, which could lead to the stabilization of these domains [65].

The cortical actin cytoskeleton, which underlies the plasma membrane, plays an important role in stabilizing microdomains [66]. In addition, increases in local F-actin concentration in response to activation signals drive the large-scale clustering of microdomains [67,68,69,70]. These functions of the actin cytoskeleton, which are important for membrane rigidity, protein organization, and lateral movement of proteins [66], are dependent on plasma membrane-associated lipids and proteins. Specialized lipids phosphatidylinositol (4,5)-bisphosphate (PIP2) and phosphatidylinositol (3,4,5)-trisphosphate (PIP3) bind to and regulate proteins involved in actin polymerization, cross-linking of filaments, and filament capping [71,72,73]. The ERM (Ezrin, Radixin and Moesin) proteins [74,75,76] and talin [77,78], all of which tether the actin cytoskeleton to the transmembrane proteins, mediate the protein-based regulation of the plasma membrane structure and function by the actin cytoskeleton. Overall, the actin cytoskeleton and the plasma membrane microdomains are intricately linked with each other.

## 4. Relationships between Virus Growth and Cytoskeleton

Various lines of evidence support multifaceted relationships between the cytoskeleton F-actin and microtubules and the IAV life cycle. First, both actin and tubulin are incorporated into released IAV particles [79,80]. Second, viral components, specifically M1 and proteins comprising the vRNPs, associate with actin and/or tubulin in host cells [81,82,83,84]. Third, IAV infection alters the levels, structures, and functions of F-actin and microtubules in host cells. Several studies show an enhancement in total actin [85,86] and tubulin [87] levels upon infection. However, one study showed no difference in cellular actin levels [88], and another showed a reduction in tubulin levels [85] upon IAV infection. In addition to total actin and tubulin levels, IAV infection can also modulate the levels or activities of proteins involved in the regulation of F-actin and microtubule dynamics. While some key proteins involved in F-actin dynamics, such as cofilin-1 [86,89], ERM proteins [86], talin [86], and the regulatory light chain of myosin II motor [86], are upregulated in response to IAV infection, others, such as Arp2/3 [88], formins [88], cortactin [90], and myosin Vb [88], are downregulated. With respect to microtubule dynamics, dynactin, which serves as a co-factor for dynein motors, is upregulated in response to IAV, while specific kinesin motors are downregulated [88]. RhoA, which regulates the functions of both microtubules and the actin cytoskeleton, was downregulated in infected epithelial cells [91]. In addition, the activity of RhoA [86,91] and another Rho GTPase, Rac1 [92], is reduced upon IAV infection but the activity of a third Rho GTPase, Cdc42 [93], is enhanced upon infection. Therefore, there seems to be some specificity with which IAV infection modulates the expression or activity of proteins engaged in F-actin and microtubule dynamics.

Fourth, the disruption of cytoskeletal dynamics has either a negative or a positive effect on viral replication. Disruption of F-actin dynamics, by drugs inhibiting polymerization (cytochalasin D), enhancing depolymerization (latrunculin A), or stabilizing actin filaments (jasplakinolide), leads to an increase in released viral titers in non-polarized cells [94,95]. In epithelial cells, the effects of these drugs on viral replication differ between different studies. In the Madin-Darby Canine Kidney (MDCK) cell line, the disruption of F-actin dynamics has no [94], a positive [95], or a negative [96,97] effect on released viral titers. In polarized rhesus monkey kidney cells, which are inefficient in supporting the infection of a laboratory adapted IAV strain, the inhibition of F-actin polymerization [98] drastically increases released viral titers. Cofilin-1 [89] and cortactin [90], which can promote the disassembly and assembly of F-actin, respectively, are both thought to play positive roles during IAV infection. With respect to microtubules, the disruption of microtubules has no [99] to a moderately negative [39,100] effect on virus propagation, whereas a reduction in efficiency of microtubule assembly increases virus titers in a rhesus monkey kidney cell line [101]. In addition, an increase in tubulin acetylation status, which promotes microtubule function in trafficking [102], was observed to correlate with an increase in virus titers in one study [103], but not in the other [99].

Among the studies listed above, studies showing infection-induced changes in cytoskeletal proteins or their association with viral proteins or particles are consistent with a role for the cytoskeleton in IAV infection. Obviously, however, studies conducted using specific inhibitors or genetic ablation should provide more definitive evidence on the role of specific cytoskeletal components in the virus life cycle. Nonetheless, some studies using these strategies show contradictory data, even as to whether the cytoskeleton of interest plays a positive role or not [39,89,94,95,96,98,99,100,101]. This could partially be attributed to the use of viral titers as readout for IAV infection. Measurement of viral titers may not allow for identification of the role(s) of the cytoskeleton at different stages of the IAV life cycle, especially if the cytoskeleton plays opposing roles in the early and late stages of IAV infection. Even with studies focusing on the late stages of the virus life cycle, the disruption of F-actin can have contradictory effects on particle assembly [94,95,96,104]. This could be because F-actin can play negative and positive roles at different steps in IAV assembly. Depending on the conditions, such as host cell types, viral strains, and the time points at which analyses are performed, cumulative effects of cytoskeletal disruption may lead to different final outcomes on virus production. Therefore, to advance our understanding of the roles played by cytoskeletons in virus growth, it is necessary to determine the effects of cytoskeleton disruption on each of the single defined steps of the virus life cycle, in addition to the overall viral titers. In the following sections of this review, we describe the individual steps of the IAV assembly process, as well as the role of different cytoskeletal components during these steps.

## 5. IAV Assembly

IAV is thought to assemble in cholesterol-enriched microdomains, or membrane rafts, of the plasma membrane of host cells [6,7,65,105,106,107]. HA and NA accumulate and co-cluster at these microdomains and form sites of virus assembly known as budozones [105,106,108,109,110,111], while the third transmembrane protein, M2, is suggested to localize at the edge of the budozone [8,106,112,113]. M1, which is predominantly cytosolic, associates with membranes containing HA, NA, and M2, either during the ER-Golgi transport or at the plasma membrane [114,115,116,117]. Expression of HA, NA, M1, and M2 in cells is sufficient to drive the assembly and budding of virus-like particles at the plasma membrane [108,118]. A subset of the M1 population is also imported into the nucleus [119,120,121,122], where it mediates the export of vRNPs via the Crm1 pathway along with another viral non-structural protein, NS2 [123,124]. After nuclear export, vRNPs are thought to co-opt the cellular recycling compartment to traffic to assembly sites at the plasma membrane [38,39,40,125]. Arrival of vRNPs at virus assembly sites further promotes the assembly and budding of virus particles [126,127,128]. While HA, NA, and likely M1 induce and stabilize membrane curvature [108,118,129], membrane scission and release of virus buds requires M2. M2 is enriched at the neck of the virus bud [8,112,113] and is thought to induce positive membrane curvature, which may be sufficient for membrane scission [8,130]. NA prevents the retention of nascent particles that have undergone the scission by cleaving cell-surface sialic acid moieties, which could otherwise bind virus-associated HA [131,132].

IAV is pleomorphic, possessing two distinct morphologies: spherical virions that are ~100 nm in diameter and filamentous particles that are ~100 nm in diameter and up to 20 um in length [133,134,135,136]. Most commonly used laboratory-adapted strains, such as A/Puerto Rico/8/1934 (H1N1) (PR8) and A/WSN/1933 (H1N1) (WSN), are solely spherical [137]; however, in vivo human infection produces both spherical and filamentous virions [133,135]. In addition, virions with a filamentous morphology emerge after the passaging of spherical viral strains in guinea pigs, suggesting the selective advantages of this morphology [136]. The key genetic determinant of virion morphology is the M1 protein, since specific mutations in the M1 protein confer the ability to form filamentous virions [136,138,139,140,141,142], although M2 [141,143,144] and NP [128] also play some roles.

F-actin and microtubules could regulate at least the following steps in the IAV assembly process: trafficking of viral transmembrane proteins to the plasma membrane, trafficking of cytoplasmic viral components to the assembly sites, association between viral components at the plasma membrane, and morphogenesis of nascent particles. In the subsequent section, we describe the roles played by the actin cytoskeleton and microtubules at these individual steps of the IAV assembly process. The different steps of IAV assembly and the roles (positive or negative) of the cytoskeleton during these steps are depicted in Figure 1. In addition, the roles of specific cytoskeletal components at different steps of IAV assembly are summarized in Table 1.

## 6. Roles Played by the Cytoskeleton at Specific Steps of IAV Assembly

### 6.1. Trafficking of Viral Transmembrane Proteins to the Apical Membrane

As mentioned above, IAV assembles at the apical surface of polarized epithelial cells, which requires the targeting of virus components to the apical plasma membrane [152,153,154,155,156,157,158,159,160,161,162,163,164]. F-actin disruption has no obvious effect on the apical targeting of transmembrane proteins HA and NA or virus assembly in polarized MDCK cells [97,145]. In one study, impairment in glycosylation of HA and NA was observed upon F-actin disruption; however, this did not have an effect on apical targeting of the proteins [97]. With respect to microtubules, earlier studies showed that their disruption reduces the apical targeting of HA in infected MDCK cells without a loss of polarity in the cell monolayer [145,146,147]. This apical transport of HA is dependent on the acetylation status of microtubules, since the deacetylation of microtubules reduces the efficiency of this transport [103]. Efficient apical targeting of HA in infected cells requires the association of a kinesin protein KIFC3 with trans Golgi-derived vesicles, further supporting the role of microtubules in this process [148]. Of note, increased mistargeting of HA to the basolateral membrane upon microtubule disruption does not inhibit virus particle assembly *per se* [145,146]; however, in one study, virus assembly was still restricted to the apical surface [145], whereas in the other, it took place at both apical and basolateral membranes [146]. More recent studies have shown that even when HA is intentionally mistargeted to the basolateral membrane, virus assembly, likely mediated by some residual apically targeted HA and/or other viral proteins, still takes place at the apical surface [165,166]. Similarly, mistargeting of M2 to the basolateral membrane still allows for particle assembly and release specifically at the apical surface, although it likely disrupts apical targeting and the incorporation of vRNPs into assembled particles [167]. Therefore, if microtubule disruption does alter the site of IAV assembly [146], it is likely due to either the mistargeting of one of the other viral components (such as NA and vRNPs) or mistargeting of multiple viral proteins, but not just HA or M2, to the basolateral membrane.

### 6.2. Trafficking of Viral Cytoplasmic Components to the Assembly Sites

In addition to viral transmembrane protein trafficking, M1 protein trafficking to the plasma membrane is also thought to at least partially rely on the ER-Golgi transport pathway, since M1 associates with HA- and NA-enriched membranes at both the ER-Golgi and the plasma membrane [114,115]. In the cytoplasm, M1 may also associate with actin, since it remains insoluble after a detergent treatment that disrupts M1-lipid interactions, but still allows M1-actin interactions [81]. However, it is not clear whether this association of M1 with F-actin mediates the trafficking of M1 to HA- and/or NA-enriched membranes or is involved in the cytoplasmic transport of the M1-vRNP complex.

Trafficking of the viral genome or vRNPs to the apical membrane is also a key step in the assembly of infectious IAV particles. Upon export from the nucleus, vRNPs associate with both the actin cytoskeleton [82] and microtubules [103,168]. While the association with F-actin is proposed to govern the intracellular localization of vRNPs [82] and partially drive their movement in the cytoplasm [169], their association with microtubules appears to be a key driver for vRNP trafficking [100,103,168,169,170]. vRNPs rely on Rab11-positive vesicles for apical targeting to assembly sites [39,40,99,170,171,172,173,174,175]. While these vesicles were thought to be derived from the ERC in earlier studies [39,40,100,127,169,171,175], a more recent study suggested that they are derived from the ER [125]. While their origin remains to be determined, the Rab11-positive vesicles, which are enriched in cholesterol [126], are proposed to promote IAV particle assembly [126,176]. Rab11-dependent transport is also important for the bundling of vRNPs so that eight unique segments can be incorporated into assembling virus particles [171,172]. The kinesin motor KIF13A is also involved in driving the Rab11-dependent apical trafficking of vRNPs [149]. While microtubule disruption reduces the association of vRNPs with Rab11 [99] and slows down their movement in the cytoplasm [170], the role of microtubules in the incorporation of vRNPs into assembling particles is not clear. Earlier studies showed a moderate reduction in infectious particle assembly upon microtubule disruption [39,100], while a more recent study showed no effect of microtubule disruption on the production of infectious virus particles [99]. The most recent understanding in the field is that intact microtubules are required for the association of vRNPs with Rab11-positive vesicles [99], but not for the bundling of vRNPs [171] or for their apical transport and incorporation into virus particles [99]. Thus, it remains to be understood how vRNPs traffic to assembly sites across the cytoplasm. A potential clue to this transport mechanism was provided by a recent study, which suggested that Rab11-dependent transport of vRNPs is mediated by the ER [125]. The sliding dynamics of ER membranes, which is dependent on microtubules resistant to nocodazole (which was used to determine the role of microtubules in vRNP trafficking [39,99,100]) [177], may conceivably drive the trafficking of vRNPs.

Overall, the findings from previous studies support roles for the actin cytoskeleton and microtubules in the apical targeting of IAV components. However, in most cases, the disruption of cytoskeletal elements has a minimal to modest effect on the overall infectious particle assembly process, suggesting that the virus may employ multiple pathways for apical targeting of its proteins.

### 6.3. Association between Viral Components at the Plasma Membrane

Viral proteins have been shown to co-cluster in microdomains of the plasma membrane [104,106,178]. However, it is not known whether these associations between viral proteins are initiated during the ER-Golgi transport or occur post-arrival at the plasma membrane. Since HA and NA accelerate each other’s apical trafficking [110], they are likely to co-traffic and associate with each other prior to their arrival at the plasma membrane. In contrast, M2 is suggested not to co-traffic with HA [110], and its association with HA (and likely NA) is a discrete step that takes place at the plasma membrane in a cell-type-dependent manner [104]. Despite this knowledge, we currently do not know the exact sequence of association between the transmembrane proteins, as well as their association with M1 and vRNPs. In addition, very little is known about the host cell mechanisms that regulate these associations.

The cortical actin cytoskeleton is likely to be involved in the regulation of co-clustering of viral transmembrane proteins through its ability to regulate the structure of plasma membrane microdomains. In fact, the concentration of cortical actin is apparently increased close to the plasma membrane in response to IAV infection, and this reorganization of F-actin is proposed to be important for efficient virus assembly and budding [89]. The cortical actin lowers the mobility of HA at the plasma membrane and enhances the clustering of HA molecules [150]. While HA movement is restricted to regions of the plasma membrane with an underlying F-actin network, the HA mobility negatively correlates with the density of the underlying F-actin [150]. A role for the actin cytoskeleton in modulating HA clustering is further supported by the presence of HA aggregates at the plasma membrane when the activity of myosin II motors is inhibited [96]. Association or co-clustering between HA with M2 is also regulated by the actin cytoskeleton. Thaa et al. used fluorescence resonance energy transfer (FRET) approaches, where fluorescent probes were fused to the cytoplasmic domains of HA and M2. They observed that cytochalasin D treatment reduces FRET between fluorescent protein fusions of HA and M2, indicating that the actin cytoskeleton plays a positive role in the association between the two proteins in the absence of any other viral proteins [151]. In an apparent contrast, we recently showed using a proximity-ligation approach that in infected primary human macrophages, which do not support efficient IAV assembly, the F-actin network suppresses the association between HA and M2. Disruption of the actin cytoskeleton with cytochalsin D restores HA-M2 association and particle assembly in macrophages, indicating that F-actin plays inhibitory roles in viral protein association/IAV assembly in these cells [104] (possible mechanisms discussed later). The discrepancies in the role of F-actin in HA-M2 association between the two above-mentioned studies might be due to differences in the cell types used in these studies (Chinese Hamster Ovary cells in [151] versus primary human blood derived macrophages in [104]). Cell-type dependent differences for the role of the F-actin and microtubules at different stages of the IAV life cycle have also been reported before [94,98,101,179,180,181]. In addition, technical differences could account for the discrepancies; for example, the presence of other viral components in infection-based experiments [104] or attachment of fluorescent proteins to the cytoplasmic domains of HA and M2 in the FRET study [151] may affect the interaction with subcortical actin. With respect to microtubules, a positive role for the microtubule-mediated transport of vRNPs on co-clustering between HA and M2 in infected HeLa cells has been described [126]. However, in that study, it is not clear whether microtubules support co-clustering between HA and M2 by changing the organization of plasma membrane microdomains directly or indirectly via cholesterol transport or by promoting the trafficking of vRNPs to the assembly sites.

### 6.4. Morphogenesis of Nascent Particles at the Plasma Membrane

In addition to regulating the association between viral proteins, the actin cytoskeleton also plays direct roles in particle morphogenesis. As described above, the formation of a spherical IAV particle is negatively regulated by F-actin in primary human macrophages. Intriguingly, spherical particle formation occurs regardless of the state of the actin cytoskeleton in a monocytic cell line that is differentiated into a macrophage-like morphology [104]. In epithelial cell lines, the effect of F-actin disruption on particle morphogenesis varies, depending on the virus morphology, with no obvious effects of F-actin disruption observed for IAV strains that assemble solely spherical particles (WSN or PR8) [95,97]. However, one caveat of previous studies with epithelial cells is that they were all performed using epithelial cell lines and not primary epithelial cells. It is becoming increasingly clear that some steps in the IAV particle assembly process are more stringently regulated in primary human cells than in cells lines [104,167], highlighting the need for more studies looking at the role of the actin cytoskeleton in primary cells. 

In the case of an IAV strain that forms filamentous particles (A/Udorn/72 [H3N2]), particle morphogenesis is highly dependent on F-actin. Both the stabilization and disruption of F-actin impede filamentous particle formation at the apical surface [94,95]. The mechanisms by which the actin cytoskeleton supports filamentous IAV assembly are not understood. The actin cytoskeleton may provide mechanical support at the base of or inside the viral filament. Of note, while proteomics studies showed the incorporation of actin molecules into released virions [79,80], F-actin has been observed to localize only at the base of the viral filament [95]. In such locations, it is possible that the actin cytoskeleton interacts with one or more viral structural proteins and drives their incorporation into the growing filamentous particle. This role of the actin cytoskeleton might not be so important for spherical particle assembly due to a requirement for the incorporation of a fewer number of copies of the structural proteins, such as HA, NA, M1, and M2, into spherical particles relative to filamentous particles [1]. Of note, the role of F-actin in particle morphogenesis has thus far been examined for only a limited number of the laboratory strains; whether the observations above hold true for the morphogenesis of a broad range of IAV strains with a varied tendency to form filamentous versus spherical particles remains to be tested.

## 7. Potential Mechanisms for F-Actin-Dependent Restriction of IAV Assembly

As described earlier, in non-permissive primary human macrophages, the actin cytoskeleton plays a negative role in the assembly of spherical (and likely filamentous) IAV particles, likely by restricting HA-M2 association at assembly sites (Figure 2A). However, the mechanism of action of the actin cytoskeleton during this step is not understood. The restrictive role of F-actin in primary human macrophages could either be due to the distinct structure and/or function of the actin network or due to the differential function or expression of additional actin-dependent host factors in this cell type. We propose three different possible mechanisms by which the actin cytoskeleton suppresses IAV assembly in macrophages:
Microdomain segregation: In this model, the actin cytoskeleton restricts the movement of HA- and M2-enriched microdomains and keeps the microdomains (and hence, HA and M2) segregated from each other. In fact, M2 is present in microdomains distinct from HA-enriched microdomains early on in the assembly process [106,109], but is later recruited to these assembly sites [112,178,182]. The cortical actin network in primary macrophages may keep these plasma membrane microdomains apart via interactions with either lipids [183,184] or cytoplasmic tails of transmembrane proteins [185,186,187]. As discussed above, the ERM proteins and talin are likely to be involved in linking the cortical actin cytoskeleton to the plasma membrane microdomains [74,76,77]. ERM-mediated tethering of transmembrane proteins can allow for these tethered proteins to form pickets, which restrict the mobility of other proteins associated with these microdomains [186,187];Suppression of membrane curvature: According to this model, the actin cytoskeleton suppresses HA- and/or NA-induced membrane curvature by modulating the plasma membrane stiffness [188,189]. This possibility is consistent with several studies that have shown that M2 is not required for the induction of membrane curvature during particle assembly [8,112,143,190] and that M2 may be recruited after the induction of membrane curvature [191,192], which is likely mediated by HA, NA, or M1. Therefore, F-actin may be modulating membrane curvature in a manner that is independent of the recruitment of M2 to assembly sites;Blocking of cytoplasmic components: In this model, the actin network inhibits IAV assembly by restricting the trafficking, incorporation, or function of additional components essential for IAV assembly, that is, M1 and vRNPs. Both M1 [81] and NP [82] are reported to associate with F-actin, and this association might suppress their mobility and trafficking to assembly sites. In addition, the dense F-actin cortex could also serve as a physical barrier to the diffusion of proteins or vesicles carrying these proteins [193,194].

These three potential mechanisms of action for the actin cytoskeleton in IAV assembly in primary human macrophages are depicted in Figure 2B. In some cases, the disruption of F-actin in epithelial cells modestly increases spherical virus production [94,95]. Therefore, it is possible that the mechanisms described above may also operate in IAV-permissive cells, depending on the condition.

## 8. Concluding Remarks and Outstanding Questions

In sum, in this review, we summarize evidence in support of multiple roles for the cytoskeleton in IAV assembly. The current evidence suggests that F-actin and microtubules can play either positive or negative roles at particular steps during IAV assembly, e.g., trafficking of viral components or protein-protein interactions. However, for most of these steps, we have yet to gain a clear understanding of the mechanisms involved. Some of the most pressing questions that remain unanswered in the field are: What is the role of the cytoskeleton in restricting IAV assembly to the apical surface of epithelial cells? What is the exact nature of the vesicular network involved in Rab11-dependent trafficking of vRNPs? Which cytoskeletal component(s) is involved in the trafficking of vRNPs to the apical membrane? What are the mechanisms by which F-actin regulates the association between HA and M2? What are the mechanisms by which F-actin promotes filamentous IAV assembly? What properties of F-actin allow it to suppress spherical particle assembly in primary human macrophages? Additional studies that address these questions using novel approaches to examine the behaviors of viral structural proteins and RNAs are likely to elucidate the exact roles of the cytoskeleton at specific steps in IAV assembly.

## Figures and Tables

**Figure 1 viruses-11-00046-f001:**
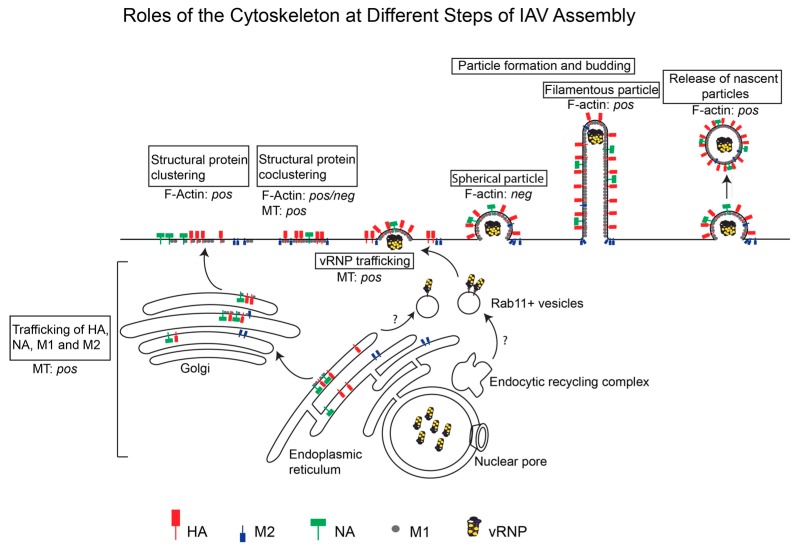
Roles of the actin cytoskeleton and microtubules at different steps of IAV Assembly. IAV assembly is initiated at the plasma membrane after the arrival of three transmembrane proteins, HA, NA, and M2, and the cytoplasmic protein M1 through the cytoplasm. Clustering of these viral proteins drives assembly of the virus particle, which could have a spherical or filamentous morphology. vRNPs are transported across the cytoplasm on Rab11+ vesicles and are incorporated into the assembling virus particle. Whether the actin cytoskeleton and/or microtubules (MT) promote (*pos*) or suppress (*neg*) individual IAV assembly steps is depicted.

**Figure 2 viruses-11-00046-f002:**
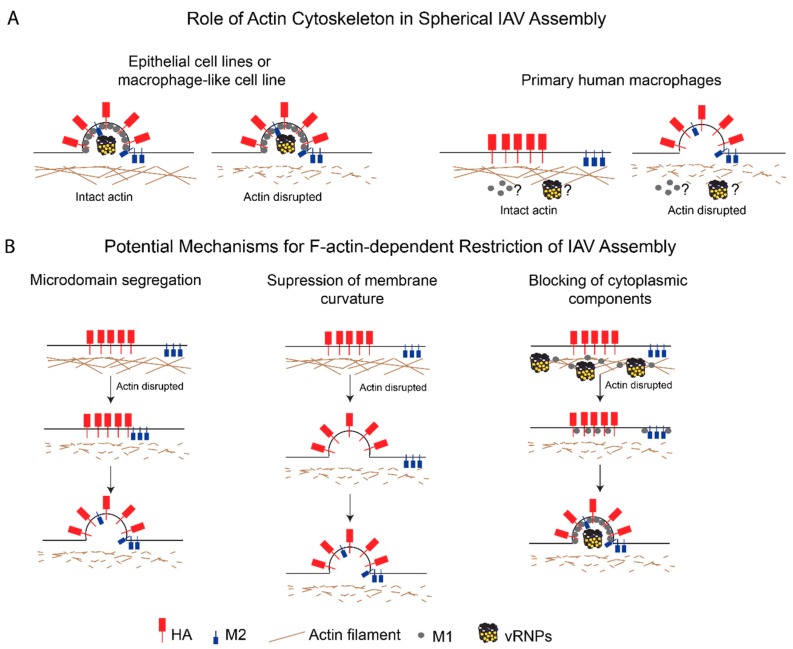
The role of the actin cytoskeleton in spherical IAV assembly in host cells. (**A**) Contrasting roles for the actin cytoskeleton in spherical IAV assembly in different cell types. In IAV-permissive cells, such as epithelial cell lines and a macrophage-like cell line, the actin cytoskeleton either promotes or has no effects on IAV particle production. In contrast, the actin cytoskeleton restricts HA-M2 association and spherical IAV assembly in primary human macrophages. (**B**) Proposed mechanisms by which actin restricts spherical IAV assembly in primary human macrophages. *Microdomain segregation*: F-actin partitions HA- and M2-enriched plasma membrane microdomains. *Suppression of membrane curvature*: F-actin restricts HA-mediated curvature induction at the plasma membrane, which is required for M2 recruitment to the assembling particle. *Blocking of cytoplasmic components*: F-actin restricts trafficking and incorporation of other curvature-inducing structural components, that is, M1 and vRNP, to virus assembly sites.

**Table 1 viruses-11-00046-t001:** Roles of specific cytoskeletal components at different steps of IAV Assembly.

IAV Assembly Step	Cytoskeletal Component	Role of Cytoskeletal Component	Reference
HA trafficking	Microtubules	Positive	[145,146,147]
	KIFC3	Positive	[148]
	Acetylated microtubules	Positive	[103]
	F-actin	No role	[97]
vRNP trafficking	Microtubules	Positive	[39,100]
	Microtubules	No role	[99]
	KIF13A	Positive	[149]
HA clustering	F-actin	Positive	[150]
	Myosin II	Positive	[96]
HA-M2 co-clustering	F-actin	Positive	[151]
	F-actin	Negative	[104]
	Microtubules	Positive	[126]
Spherical particle assembly	F-actin	Negative	[104]
	F-actin	No role	[95,97,104]
Filamentous particle assembly	F-actin	Positive	[94,95]
Release of nascent particles	F-actin	Positive	[97]
	F-actin	No role	[94,95,104]

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
