# Peer review of "Friend or Foe: The Role of the Cytoskeleton in Influenza A Virus Assembly"

_viruses, 2019, doi:10.3390/v11010046_

Reviewer 1 Report

GENERAL COMMENTS:
The review on the role of the cytoskeleton in IAV infection does a reasonable job in covering the literature. A number of points listed below should be addressed to provide greater clarity, substance and flow within the review.

SPECIFIC COMMENTS:
1. The section of "4. IAV assembly" is in an awkward place. As it talks about the general process of IAV assembly, it might be better placed before section 6.

2. A table should be included summarising the cytoskeleton proteins and their mode of action relevant to Fig. 1. This would complement the figure and provide greater clarity.

3. As IAV has many strains, is there any strain-specific? For example, the shape of the viral particles is strain-dependent in the in vitro culture system.

Author Response

We thank the reviewer for the suggestions, which we incorporated in this revision as following.

1. The section of "4. IAV assembly" is in an awkward place. As it talks about the general process of IAV assembly, it might be better placed before section 6.

This section is now placed before Section 6. Accordingly, new transition sentences have been added.

2. A table should be included summarising the cytoskeleton proteins and their mode of action relevant to Fig. 1. This would complement the figure and provide greater clarity.

We have included a table as requested by the reviewer. To improve the correlation between the table and Figure 1, we also modified the figure and added the step of “Release of nascent particles”.

3. As IAV has many strains, is there any strain­specific? For example, the shape of the viral particles is strain­dependent in the in vitro culture system.

In response to this comment, we modified the section 6.4 “Morphogenesis of nascent particles at the plasma membrane”. Specifically, we clarified which strains have been examined in the in vitro experiments for the roles played by cytoskeletons and pointed out that the studies on the role of cytoskeletons in virus morphogenesis have been done only for a limited number of lab strains.   

Reviewer 2 Report

The review by Sukhmani Bedi and Akira Ono is a very well written, well organized and nice overview of the role of the cytoskeleton in influenza A virus assembly. This manuscript introduced the life cycle of influenz virus, overview of the cytoskeleton, cytoskeletal functions relevant to IAV infection, and IAV assembly, then further highlighted the role of cytoskeletal elements in cellular processes that are associated with influenza virus in the host cell from the point of cell biology. Finally, the questions that remain unanswered in this field were suggested, providing some meaningful directions for the future study. problem The review is nice to read and is very documented and complete. I think it deserves publication, as it represents a useful information for many scientists working in the field. Please note that there are two 3.3 in the text which should be corrected.

Author Response

We thank the reviewer for the very positive comments. We have corrected the error pointed out by the reviewer.